computational chemistry/organic chemistry/ physical chemistry

aromaticity, nitrogen confused porphyrin, topological resonance energy, ring current

**Author for correspondence:**
Ablikim Kerim
e-mail: ablikim.kerim@163.com

This article has been edited by the Royal Society of Chemistry, including the commissioning, peer review process and editorial aspects up to the point of acceptance.

# A study on the aromaticity and magnetic properties of N-confused porphyrins

## Maimaitijiang Tuersun and Ablikim Kerim

The College of Chemistry and Chemical Engineering, Xinjiang University, Urumqi 830046, People's Republic of China

AK, 0000-0002-4621-8692

In this paper, topological resonance energy (TRE) methods were used to describe the global aromaticity of nitrogen confused porphyrin (NCP) isomers. The TRE results show that all NCP isomers exhibit lower aromaticity than the normal porphyrins, and their aromaticity decreases as the number of confused pyrrole rings in the molecule increases. In the NCPs, global aromaticity decreases as the distance between the nitrogen atoms increases. The bond resonance energy (BRE) and circuit resonance energy (CRE) indices were applied to study local aromaticity and conjugated pathways. Both the BRE and CRE indices revealed that individual pyrrolic subunits maintain their strong aromatic character and are the main source of global aromaticity. Ring currents (RC) were analysed using the Hückel–London model. RC results revealed that the macrocyclic electron conjugation pathway is the main source of diatropicity. As the number of confused pyrrole rings in the molecule increases, its diatropicity gradually decreases. In the confused pyrrole rings of the NCP isomers, the diatropic RC passing through the $\beta$-positions is always weaker than that passing through the inner sections. This is unrelated to the location of the protonated or non-protonated nitrogen atom at the periphery of the molecule and must be ascribed to the unique properties of the confused pyrrole rings.

# 1. Introduction

The aromaticity and main conjugation pathways of porphyrins and the related macrocycle have been subjects of considerable interest for many years [1–5]. N-confused porphyrins (NCPs) are a class of porphyrin isomers in which one or more of the core nitrogen atoms faces outward from the macrocycle and a CH unit faces inward toward the core [6,7]. To be specific, one or more pyrrole rings are connected to the *meso* carbon atoms at the $\alpha$ and $\beta$ positions. Many reports have demonstrated that these compounds possess properties quite different from normal porphyrins in terms of their aromatic character and in their

ability to stabilize metals in unusual states of oxidation [8,9]. Thus, research on NCPs has also received considerable attention [10]. One of the important properties of NCPs is the hydrogen atom shifts between the inner and peripheral nitrogen atoms in the confused pyrrole ring which generate two types of tautomers and, under mild conditions, often result in substantial changes in their aromaticity [11]. This structural change in NCPs also leads to the existence of physical and chemical properties different from those observed in normal porphyrins. Compared with the progress in understanding normal porphyrins, the chemistry of NCPs is under-explored [12,13]. From a theoretical viewpoint, the stability, structure and aromaticity of 95 types of NCP isomers have been investigated by Furuta et al. [6] using density functional theory (DFT) calculations. However, the local aromaticity of these isomers remains unclear, and yet to be studied is whether or not 18π-electron delocalization is the main source of both the aromaticity and magnetropicity of these molecules as a whole.

Aromaticity is one of the most important general concepts in physical organic chemistry and has been widely used for interpreting the stability of a molecule [14–16]. Aromaticity is predicted on the basis of the energetic, structural, magnetic and electronic properties of a given system [17–22]. Of these approaches, the topological resonance energy (TRE) method is one of the most reliable energetic determinants of aromaticity and has been used to solve many different problems relating to aromaticity [14–16]. Many indices of local aromaticity have been proposed for characterizing individual rings in polycyclic compounds [23,24]. Bond resonance energy (BRE) and circuit resonance energy (CRE) indices are the simplest way to estimate the relative local aromaticities in polycyclic compounds such as porphyrins [16,25–30].

In the current work, we report the global aromaticity of NCP isomers and compare these with the normal porphyrins by the use of the TRE method. Our results are then compared to the trends in stability predicted by Furuta et al. [6] who used the DFT method. We discuss the local aromaticities and the main conjugation pathways of the NCP isomers as determined using two metrics, the BRE and the CRE indices. The ring current (RC) strength results we obtained are discussed and compared to experimental data.

## 2. Methods of calculation

TRE is associated with the degree of π-electron delocalization in cyclic conjugated systems [14,15,31,32]. The greater the TRE values, the more stable a molecule is and the higher its aromatic character. BRE represents the contribution of a given π-bond to the TRE [33]. If the BRE is a positive value, this indicates that a stabilizing contribution is made by a given π-bond to the molecule as a whole; if the BRE is a negative value, a destabilizing contribution is indicated [34]. The CRE value can be used to measure the contribution of resonance energy that each circuit makes to the TRE [35]. BRE and CRE provide information about each chemical bond and each individual ring, respectively [25–30]. TRE, BRE and CRE are given in units of $|\beta|$, where $\beta$ is the standard resonance integral in Hückel theory. RC strength herein has been obtained using the Hückel–London model [36–41]. Van-Catledge's set of Hückel parameters have been used for nitrogen [42]. The TRE, BRE, CRE and RC indices can be defined graph-theoretically within the framework of simple Hückel molecular orbital theory [35].

## 3. Results and discussion

### 3.1. Global aromaticity

In 2001, Furuta et al. [6]. reported the structure and aromaticity of 95 kinds of porphyrin isomers, from single N-confused to multiple N-confused isomers. In the present study, we have considered 37 of these isomers. The remaining isomers were excluded since they represent non-conjugated systems. The systems chosen for this study are given in figure 1. For the convenience of discussion, in this study, the compounds are named according to the number of confused nitrogen atoms in the parent molecule and have been classified into four types: normal porphyrin ($N_0CP$), singly N-confused porphyrin ($N_1CP$), doubly N-confused porphyrin ($N_2CP$) and triply N-confused porphyrin ($N_3CP$). The molecules are named as they were named in the Furuta et al. [6]. The TREs are presented in table 1. The nucleus-independent chemical shift (NICS) index is one of the most widely used magnetic criteria of aromaticity. The index is defined as the negative value of the absolute magnetic shielding computed at ring centre NICS(0), and 1 Å above the centre of the ring NICS(1) and its zz-tensor component NICS(1)$_{zz}$, where the z-axis is a normal to the ring plane [43–48]. The relative energies

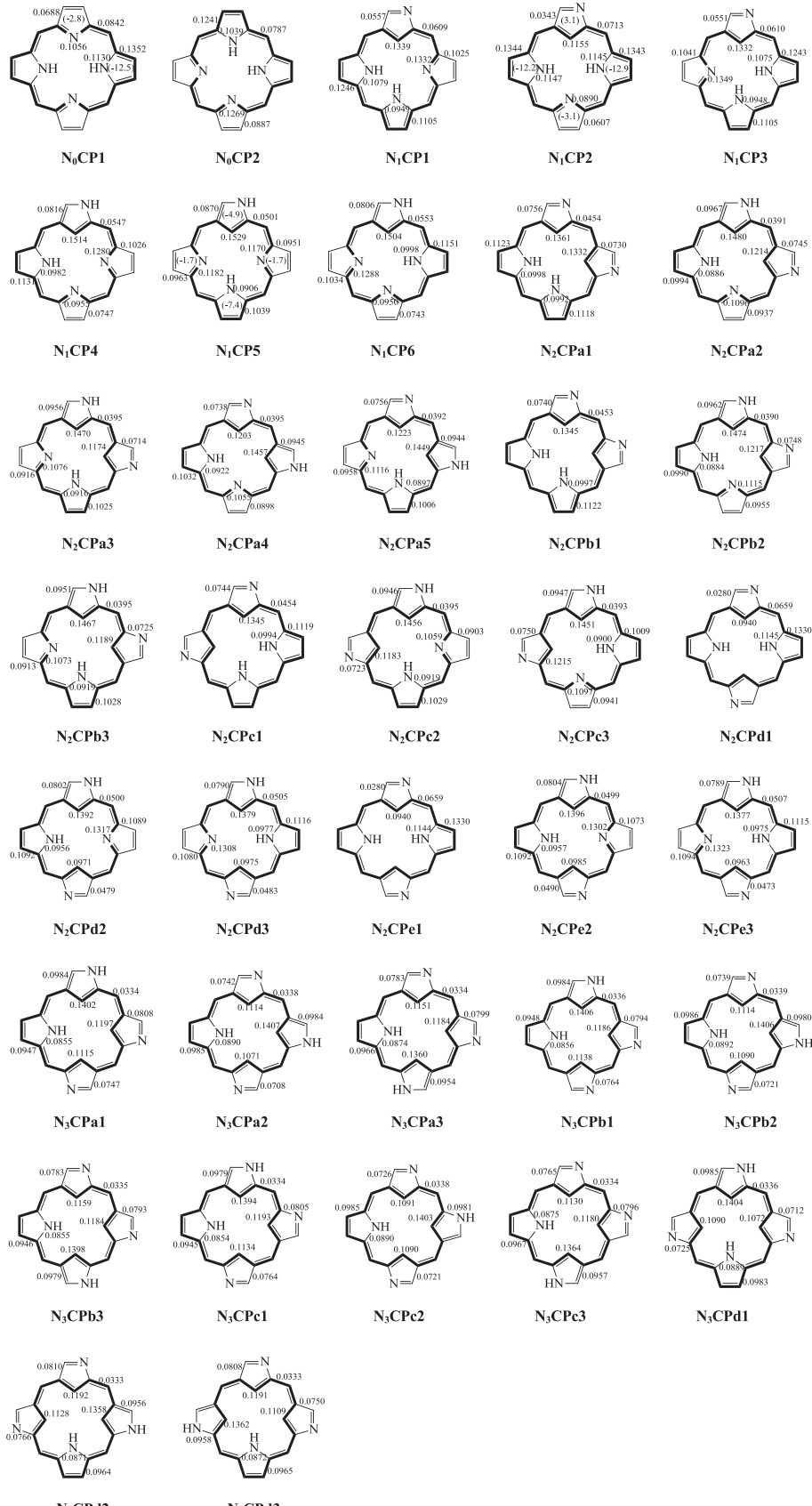

**Figure 1.** The structures of the NCPs in this study. The BRE values are in units of $|\beta|$ for all $\pi$ bonds. The main aromatic conjugation pathways are shown in bold. The NICS(0) values for the individual rings of **N₀CP1**, **N₁CP2** and **N₁CP5** are given in parentheses. These values are taken from reference [54].

**Table 1.** Global aromaticity indices of normal porphyrins and of NCP isomers.

| species | TRE | TRE$_{mean}$ | %TRE | MRE | NICS | NICS(0)$_{mean}$ | R.E. (kcal mol$^{-1}$) |
|---|---|---|---|---|---|---|---|
| N$_0$CP1 | 0.4322 | | 1.1699 | 0.3390 | −15.12 | | 0 |
| N$_0$CP2 | 0.4476 | 0.4399 | 1.2116 | 0.3553 | −14.89 | −15.01 | 8.185 |
| N$_1$CP1 | 0.4366 | | 1.1828 | 0.3555 | −13.15 | | 23.631 |
| N$_1$CP2 | 0.4074 | | 1.1037 | 0.3309 | −13.84 | | 17.147 |
| N$_1$CP3 | 0.4375 | | 1.1853 | 0.3594 | −13.05 | | 23.300 |
| N$_1$CP4 | 0.4393 | 0.4312 | 1.2322 | 0.3714 | −7.71 | −10.41 | 32.107 |
| N$_1$CP5 | 0.4394 | | 1.1945 | 0.3685 | −7.03 | | 24.928 |
| N$_1$CP6 | 0.4267 | | 1.1602 | 0.3602 | −7.7 | | 31.683 |
| N$_2$CPa1 | 0.4312 | | 1.1688 | 0.3641 | −11.02 | | 43.127 |
| N$_2$CPa2 | 0.4248 | | 1.1557 | 0.3687 | −5.28 | | 41.121 |
| N$_2$CPa3 | 0.4251 | | 1.1566 | 0.3657 | −5.51 | | 39.392 |
| N$_2$CPa4 | 0.4252 | | 1.1569 | 0.3657 | −3.84 | | 37.615 |
| N$_2$CPa5 | 0.429 | | 1.1671 | 0.3688 | −5.05 | | 38.569 |
| N$_2$CPb1 | 0.4308 | | 1.1677 | 0.3624 | −11.22 | | 43.352 |
| N$_2$CPb2 | 0.4272 | 0.4176 | 1.1621 | 0.3695 | −4.34 | −7.61 | 38.744 |
| N$_2$CPb3 | 0.4263 | | 1.1599 | 0.3664 | −4.28 | | 37.461 |
| N$_2$CPc1 | 0.4312 | | 1.1688 | 0.3640 | −11.29 | | 43.504 |
| N$_2$CPc2 | 0.4231 | | 1.1511 | 0.3644 | −4.66 | | 39.324 |
| N$_2$CPc3 | 0.4257 | | 1.1581 | 0.3674 | −4.81 | | 40.640 |
| N$_2$CPd1 | 0.3812 | | 1.0334 | 0.3164 | −13.72 | | 37.524 |
| N$_2$CPd2 | 0.4087 | | 1.1124 | 0.3535 | −8.37 | | 43.612 |
| N$_2$CPd3 | 0.4103 | | 1.1167 | 0.3530 | −7 | | 42.239 |
| N$_2$CPe1 | 0.3812 | | 1.0335 | 0.3164 | −13.63 | | 37.503 |
| N$_2$CPe2 | 0.4087 | | 1.1125 | 0.3539 | −7.36 | | 42.876 |
| N$_2$CPe3 | 0.4102 | | 1.1165 | 0.3528 | −7.94 | | 43.141 |
| N$_3$CPa1 | 0.4150 | | 1.1303 | 0.3657 | −6.92 | | 59.357 |
| N$_3$CPa2 | 0.4111 | | 1.1197 | 0.3601 | −7.26 | | 55.283 |
| N$_3$CPa3 | 0.4184 | | 1.1396 | 0.3656 | −5.96 | | 54.871 |
| N$_3$CPb1 | 0.4159 | 0.4144 | 1.1329 | 0.3666 | −3.94 | −6.32 | 56.101 |
| N$_3$CPb2 | 0.4127 | | 1.1241 | 0.3610 | −4.42 | | 54.452 |
| N$_3$CPb3 | 0.4179 | | 1.1385 | 0.3770 | −5.83 | | 56.353 |
| N$_3$CPc1 | 0.4166 | | 1.1346 | 0.3659 | −5.91 | | 57.027 |
| N$_3$CPc2 | 0.4102 | | 1.1172 | 0.3595 | −6.59 | | 54.720 |
| N$_3$CPc3 | 0.4154 | | 1.1315 | 0.3645 | −6.54 | | 56.668 |
| N$_3$CPd1 | 0.4086 | | 1.1127 | 0.3585 | −9.33 | | 56.711 |
| N$_3$CPd2 | 0.4169 | | 1.1354 | 0.3645 | −5.73 | | 56.713 |
| N$_3$CPd3 | 0.4144 | | 1.1285 | 0.3637 | −7.52 | | 57.644 |

(R.E.) and NICS(0) values of the macrocyle were calculated by the GIAO method at the B3LYP/6–31G$^{**}$ level by Furuta *et al.* [6]. The NICS(0) and the R.E. values reported for these compounds are also presented in table 1. These were used by Furuta *et al.* as global aromaticity indices for these compounds. According to the results of the TRE calculations shown in table 1, all of these isomers are predicted to be aromatic with positive TRE values. We can define the TRE$_{mean}$ and the NICS(0)$_{mean}$ as the mean TRE and NICS(0) value, respectively, for the same group of isomers. By comparing the

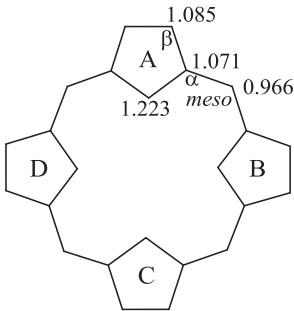

**Figure 2.** The URFs for normal porphyrins and of NCP isomers. The $\alpha$, $\beta$ and *meso* positions are indicated on the URF. In normal porphyrins and in the NCP isomers in this study, the four types of rings are denoted by A, B, C and D.

trend of the $\text{TRE}_{\text{mean}}$ results for the four types of isomers, we found that as the number of the *confused* pyrrole rings in a molecule increases, the $\text{TRE}_{\text{mean}}$ values gradually and continually decrease. Thus the pattern is

$$\text{N}_0\text{CP} > \text{N}_1\text{CP} > \text{N}_2\text{CP} > \text{N}_3\text{CP}.$$

The above aromatic trend as predicted by the $\text{TRE}_{\text{mean}}$ method is good in agreement with the $\text{NICS}(0)_{\text{mean}}$ and R.E. results obtained by Furuta *et al.* using the DFT method. Generally speaking, aromaticity reflects the stability of a molecule. For these isomers, the trends in stability and aromaticity exhibit rough positive correlation. There were, however, some compounds where the TRE and R.E., as well as the NICS(0), showed no trend, and no correlation could be found between them [49].

The relative aromaticity of these compounds may also be explained in terms of the topological charge stabilization (TCS) rule [50,51]. According to the TCS rule, the best placement of electronegative heteroatoms is at those positions with the greatest charge within the uniform reference frame (URF) [50,51]. The URF is given in figure 2. From it, we can see that the inner position of the five-membered ring exhibits high charge density. According to the TCS rule, the nitrogen atoms, with larger electronegativity, should prefer to be located at the inner position of the five-membered rings in order to stabilize these molecules. For $\text{N}_0\text{CP1}$ and $\text{N}_0\text{CP2}$, four electronegative nitrogen atoms are located at positions which have the highest charge density (1.223) in the URF. Thus, $\text{N}_0\text{CP1}$ and $\text{N}_0\text{CP2}$ obey the TCS rule as a whole. However, in the NCP isomers, at least one nitrogen atom occupies the position of the second highest charge density (1.085) in the URF. This is in opposition to the TCS rule and causes the stability of the NCP molecules to decrease relative to the $\text{N}_0\text{CP1}$ and $\text{N}_0\text{CP2}$ isomers. As the number of the confused pyrrole rings in a molecule increases, the disadvantageous factors also increase. Thus, the $\text{N}_3\text{CP}$ isomers exhibit the least aromaticity.

## 3.2. Local aromaticity and the main conjugation pathways

In order to obtain deeper insight into the aromatic properties of these compounds, we investigated local aromaticity by calculating the BRE values of various chemical bonds and the CRE values of various circuits. Based on those BRE and CRE values, we were able to predict the main conjugation pathways of these compounds.

### 3.2.1. BRE results

The BRE values for all non-identical $\pi$-bonds are included in figure 1. As shown there, all bonds in these compounds exhibit positive BRE values, meaning that they all make an aromatic contribution to the molecule as a whole. By comparing the BRE values of $\text{N}_0\text{CP1}$ with $\text{N}_0\text{CP2}$, we found that in the protonated pyrrole rings, the C–C bonds exhibit larger BRE values than the C–N bonds. However, in the non-protonated pyrrole rings, the C–C bonds exhibit smaller BRE values than the C–N bonds. In the NCP isomers, all pyrrole ring bonds exhibit relatively large positive values similar to those in $\text{N}_0\text{CP1}$ and $\text{N}_0\text{CP2}$, and these rings retain their relatively large BRE values. However, the $\text{C}_\alpha$–$\text{C}_{\text{meso}}$ bonds exhibit smaller BRE values than those in both the protonated pyrrole rings and the non-protonated pyrrole rings. With an increase in the number of *confused* pyrrole rings in the molecule,

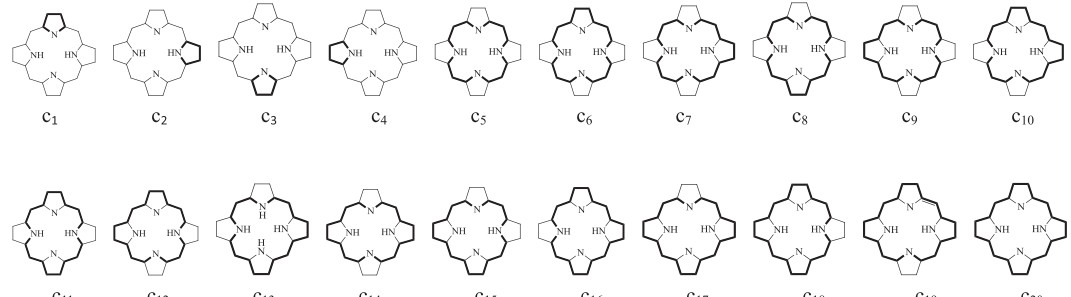

**Figure 3.** The circuits in normal porphyrins and NCP isomers.

the BRE values of the $C_\alpha$–$C_{meso}$ bonds gradually decrease. However, the BRE values of the *confused* pyrrolic ring bonds remain essentially the same. From the BRE values in the different rings in all these compounds, we can conclude that the four pyrrolic rings are the main source of aromaticity. The *meso*-position exhibits lower local aromaticity than any other position in the ring. Generally speaking, the least aromatic section in the polycyclic molecule is the active position [52]. In fact, the bromination of $N_0CP1$ occurs at the *meso*-position [53]. Our prediction of the BRE results is in agreement with the experimental observations that have been reported for $N_0CP1$ [53].

In order to further understand the effects of nitrogen atoms on global aromaticity, we have studied the aromaticity of subunits of the NCP isomers. These structures are shown in electronic supplementary material, figure S1. The BRE values of these compounds are presented in electronic supplementary material, figure S1 of the supporting information. The TRE values are given in electronic supplementary material, table S1. By comparing the TRE and %TRE values in electronic supplementary material, table S1, we found that in these compounds, the protonated and the non-protonated nitrogen atoms which are separated by three carbon atom structures exhibit larger TRE and %TRE values than the two nitrogen atoms which are separated by four carbon atom structures. From the BRE values in electronic supplementary material, figure S1, we found that the local aromaticity of the protonated pyrrole ring is unchanged. The local aromaticity of the non-protonated pyrrole ring, however, decreases as the distance between the two nitrogen atoms increases. From this, we can conclude that the relative aromaticity of NCP isomers strongly depends on the position of the two aza points. In $N_0CP1$ and $N_0CP2$, two nitrogen atoms are separated by three carbon atoms. Thus, in terms of their ability to attract electrons, the four nitrogen atoms cancel each other out, which is favourable for the uniform distribution of electrons in the molecule and results in greater delocalization of the electrons. In the NCP isomers, if one of the *β*-position carbon atoms is replaced by a nitrogen atom, the newly introduced nitrogen atom is separated from the three or four inner nitrogen atoms. The distance between these groups of nitrogen atoms is increased and less delocalization of the substructure is the result. Thus, in the NCP isomers, the *β*-position exhibits gradually decreasing local aromaticity as the number of confused pyrrole rings increases. Decreasing the local aromaticity at the four *meso*-positions plays a crucial role in the reduction of the global aromaticity of the NCP isomers. Another way to express this is to say that the stabilities of these structures are strongly affected by the position of the aza points. The farther the nitrogens are from one another, the less aromatic the system is. In general, a five-membered ring in a polycyclic system tends to attract approximately six π electrons to it. Thus, non-protonated nitrogen atoms must be somewhat similar in π-bond character to protonated nitrogen atoms. Still, from an analysis of the BRE values in NCP isomers, it is difficult to determine the differences in local aromaticity between the protonated and the non-protonated pyrroles. In most of the cases, protonated pyrrole rings exhibit larger BRE values than do the non-protonated pyrrole rings. At the same time, in NCP isomers all confused pyrrolic rings exhibit very similar local aromaticity to that of the normal pyrrolic rings within that same molecule.

### 3.2.2. CRE results

In order to further understand the contributions of the individual circuits to global aromaticity, and to predict the annulene-type conjugation pathway, we have also calculated the CRE values of the above compounds. As shown in figure 3, the π-electron ring system of these compounds contains 20 circuits

which we have labelled from $c_1$ to $c_{20}$. The $c_1$ to the $c_4$ circuit values are related to four pyrrolic rings. The values of the other 16 circuits are related to the macrocycle that encloses the inner cavity. The CRE values of the compounds are listed in table 2. As shown in table 2, the CRE values calculated for the $c_1$, $c_2$, $c_3$ and $c_4$ circuits of all compounds are comparatively large. However, the 16 circuits labelled $c_5$ to $c_{20}$ exhibit markedly lower CRE values. This demonstrates that the four constituent pyrrolic rings are the main origin of local aromaticity and they dominate the aromaticity of the molecule as a whole. In $N_0CP1$ and $N_0CP2$, the protonated pyrrolic rings exhibit greater electron delocalization than the pyrrolic rings without protonated nitrogen. With regard to the NCP isomers, the changes in their CRE values are similar to those in $N_0CP1$ and $N_0CP2$.

The NCPs can be divided into two groups depending on the location of the nitrogen atom(s) in relation to the periphery of the ring. In the first group of isomers, the two protonated nitrogen atoms are located inside the ring and one or two non-protonated nitrogen atoms are located on the periphery of the ring. In the second group of NCP isomers, however, one of the protonated nitrogen atoms is located on the periphery of the ring. Main macrocyclic circuits, such as an aromatic $18\pi$-system in free base porphine, are called main macrocyclic annulene pathways. Furuta *et al.* [6] predicted the macrocyclic annulene pathways of $N_0CP1$, $N_0CP2$ and of our first group of isomers, and these are shown in bold in figure 1. The CRE values are useful for identifying the main macrocyclic annulene pathways in porphyrins [25,27–30]. In the macrocyclic annulene pathways, the circuit located along the $18\pi$ annulene ring corresponds to the largest CRE value. For example, as shown in table 2, circuit $c_{14}$ in $N_0CP1$ and circuit $c_{10}$ in $N_0CP2$, which lie along the main conjugation pathway, are the ones we selected because they have the largest positive CRE among the macrocyclic circuits. Using the same approach, we have also predicted the main macrocyclic conjugation pathways of the first group of isomers. For these compounds, our predictions of the macrocyclic conjugation pathways using the CRE values are identical with the results predicted by Furuta *et al.* [6]. BRE values can also be used to determine the macrocyclic annulene pathways of porphyrins [25–30]. In the first group of isomers, all $\pi$-bonds located in the [18] annulene or $18\pi$-electron conjugation pathways are intensified with larger positive BREs than those located along the bypasses. The macrocyclic annulene pathways of our second group of compounds were not determined by Furuta *et al.* [6]. When we use the same method to predict the macrocyclic conjugation pathways for this second group of compounds, the formally $17\pi$ annulene rings are shown to have the largest BRE and CRE values. The macrocyclic conjugation pathways of these compounds are also shown in bold in figure 1. The prediction of the macrocyclic conjugation pathways of these types of compounds is still in its infancy. Because of this, there is some suspicion that these may not satisfy Hückel's ($4n + 2$) rule of aromaticity, but that they are in fact aromatic. In these compounds, a hydrogen atom shifts between the inner and outer nitrogen sites. For example, the proton migration in compound $N_1CP2$ can cause the compound to convert to $N_1CP5$ and it can exist in two tautomeric forms [11]. The rapid tautomerism of these isomers can be found both in solution and solids. As shown in table 1, the TRE and %TRE differences between the $N_1CP2$ and $N_1CP5$ are quite small. According to the CRE values in table 2, we can expect that it is this proton migration which induces small changes in the CRE values. Based upon the CRE and BRE values, we can predict that the main macrocyclic annulene pathways of the $N_0CP$ and of the first group of isomers will always be through the nitrogen atoms of the non-protonated pyrroles and through the $\beta$-position of protonated pyrrole rings. However, in the second group of isomers, the formally main macrocyclic annulene pathways follow the inner section and never select the $\beta$-positions of the confused pyrrole rings. Traditionally, porphyrins have been described compounds within which $18\pi$ conjugation pathways make the dominant contribution to aromaticity. If the $18\pi$ conjugation pathways are truly the sole or the primary contributors in determining the aromaticity of porphyrins, the second group of isomers should show much lower aromaticity than the first group of isomers. The reason is that in the second group of isomers, the $18\pi$ conjugation pathways are formally disrupted by the protonated nitrogen atom in the confused pyrrole ring which produces a formally $17\pi$ conjugation pathway. However, the TRE and %TRE values of the second group of isomers is close to the values of the first group of isomers. This proves again that the $18\pi$ macrocyclic conjugation pathway is not the main contributor to the global aromaticity of these compounds.

The sum of the CRE values of all $\pi$-electron systems is expressed by MRE. It can be seen from the calculations that TRE and MRE here have very high correlation coefficients, equalling 0.9780. Thus, the global aromaticity may be regarded as the total contribution of all sources of local aromaticity in the individual rings of the molecule.

Marchand *et al.* [54] investigated the global and local aromaticity of the three compounds $N_0CP1$, $N_1CP2$ and $N_1CP5$, using NICS(0) criteria. The NICS(0) values reported by Marchand

**Table 2.** CRE values of normal porphyrins and of NCP isomers.

| | $N_0CP1$ | $N_0CP2$ | $N_1CP1$ | $N_1CP2$ | $N_1CP3$ | $N_1CP4$ | $N_1CP5$ | $N_1CP6$ | $N_2CPa1$ | $N_2CPa2$ | $N_2CPa3$ | $N_2CPa4$ | $N_2CPa5$ |
|---|---|---|---|---|---|---|---|---|---|---|---|---|---|
| $c_1$ | 0.0571 | 0.0727 | 0.0689 | 0.0523 | 0.0683 | 0.0844 | 0.0880 | 0.0836 | 0.0837 | 0.0953 | 0.0937 | 0.0803 | 0.0804 |
| $c_2$ | 0.0780 | 0.0727 | 0.0862 | 0.0830 | 0.0807 | 0.0866 | 0.0832 | 0.0767 | 0.0819 | 0.0809 | 0.0800 | 0.0930 | 0.0937 |
| $c_3$ | 0.0571 | 0.0724 | 0.0709 | 0.0533 | 0.0708 | 0.0672 | 0.0707 | 0.0664 | 0.0790 | 0.0854 | 0.0746 | 0.0818 | 0.0733 |
| $c_4$ | 0.0780 | 0.0724 | 0.0810 | 0.0832 | 0.0871 | 0.0752 | 0.0828 | 0.0861 | 0.0794 | 0.0721 | 0.0820 | 0.0751 | 0.0863 |
| $c_5$ | 0.0049 | 0.0047 | 0.0054 | 0.0062 | 0.0054 | 0.0083 | 0.0076 | 0.0083 | 0.0058 | 0.0085 | 0.0085 | 0.0085 | 0.0084 |
| $c_6$ | 0.0022 | 0.0078 | −0.0001 | −0.0001 | −0.0001 | 0.0001 | −0.0001 | −0.0001 | 0.0001 | 0.0002 | 0.0002 | 0.0005 | 0.0005 |
| $c_7$ | 0.0082 | 0.0078 | −0.0011 | 0.0103 | 0.0091 | 0.0042 | 0.0039 | 0.0144 | 0.0001 | 0.0004 | 0.0005 | 0.0002 | 0.0002 |
| $c_8$ | 0.0022 | 0.0021 | 0.0091 | 0.0029 | 0.0091 | 0.0042 | 0.0132 | 0.0042 | 0.0100 | 0.0046 | 0.0151 | 0.0046 | 0.0149 |
| $c_9$ | 0.0082 | 0.0021 | 0.0091 | 0.0103 | 0.0026 | 0.0142 | 0.0039 | 0.0042 | 0.0100 | 0.0150 | 0.0046 | 0.0150 | 0.0046 |
| $c_{10}$ | 0.0035 | 0.0125 | −0.0002 | −0.0004 | −0.0003 | −0.0002 | −0.0002 | −0.0004 | −0.0005 | −0.0002 | −0.0002 | −0.0002 | −0.0002 |
| $c_{11}$ | 0.0009 | 0.0033 | −0.0003 | −0.0002 | −0.0003 | −0.0002 | −0.0004 | −0.0003 | −0.0001 | −0.0002 | 0.0000 | 0.0000 | 0.0004 |
| $c_{12}$ | 0.0035 | 0.0033 | −0.0003 | −0.0004 | −0.0002 | −0.0040 | −0.0002 | −0.0003 | −0.0001 | −0.0001 | −0.0001 | 0.0004 | 0.0000 |
| $c_{13}$ | 0.0035 | 0.0033 | 0.0041 | 0.0046 | 0.0149 | 0.0019 | 0.0065 | 0.0070 | −0.0001 | 0.0000 | 0.0004 | −0.0001 | 0.0000 |
| $c_{14}$ | 0.0131 | 0.0033 | 0.0041 | 0.0169 | 0.0041 | 0.0069 | 0.0018 | 0.0070 | −0.0001 | 0.0004 | 0.0000 | 0.0000 | −0.0001 |
| $c_{15}$ | 0.0035 | 0.0008 | 0.0149 | 0.0046 | 0.0041 | 0.0069 | 0.0065 | 0.0019 | 0.0169 | 0.0077 | 0.0078 | 0.0078 | 0.0077 |
| $c_{16}$ | 0.0013 | 0.0051 | −0.0004 | −0.0005 | −0.0009 | −0.0003 | −0.0005 | −0.0005 | −0.0002 | −0.0001 | −0.0004 | −0.0001 | −0.0004 |
| $c_{17}$ | 0.0053 | 0.0013 | 0.0064 | 0.0072 | 0.0064 | 0.0029 | 0.0028 | 0.0030 | −0.0005 | −0.0002 | −0.0002 | −0.0004 | −0.0004 |
| $c_{18}$ | 0.0013 | 0.0013 | −0.0009 | −0.0005 | −0.0004 | −0.0005 | −0.0005 | −0.0002 | −0.0005 | −0.0004 | −0.0004 | −0.0002 | −0.0002 |
| $c_{19}$ | 0.0053 | 0.0051 | −0.0004 | −0.0011 | −0.0004 | −0.0005 | −0.0002 | −0.0005 | −0.0002 | −0.0004 | −0.0001 | −0.0004 | −0.0001 |
| $c_{20}$ | 0.0018 | 0.0017 | −0.0008 | −0.0009 | −0.0008 | −0.0003 | −0.0003 | −0.0004 | −0.0004 | −0.0002 | −0.0002 | −0.0002 | −0.0002 |

| | $N_2CPb1$ | $N_2CPb2$ | $N_2CPb3$ | $N_2CPc1$ | $N_2CPc2$ | $N_2CPc3$ | $N_2CPd1$ | $N_2CPd2$ | $N_2CPd3$ | $N_2CPe1$ | $N_2CPe2$ | $N_2CPe3$ | $N_3CPa1$ |
|---|---|---|---|---|---|---|---|---|---|---|---|---|---|
| $c_1$ | 0.0828 | 0.0951 | 0.0935 | 0.0826 | 0.0929 | 0.0936 | 0.0460 | 0.0825 | 0.0816 | 0.0460 | 0.0828 | 0.0814 | 0.0962 |
| $c_2$ | 0.0828 | 0.0810 | 0.0809 | 0.0791 | 0.0821 | 0.0735 | 0.0851 | 0.0918 | 0.0768 | 0.0850 | 0.0909 | 0.0766 | 0.0848 |
| $c_3$ | 0.0782 | 0.0864 | 0.0749 | 0.0791 | 0.0749 | 0.0853 | 0.0460 | 0.0615 | 0.0612 | 0.0460 | 0.0625 | 0.0603 | 0.0833 |
| $c_4$ | 0.0782 | 0.0719 | 0.0817 | 0.0826 | 0.0791 | 0.0798 | 0.0851 | 0.0749 | 0.0904 | 0.0852 | 0.0750 | 0.0912 | 0.0712 |
| $c_5$ | 0.0057 | 0.0085 | 0.0085 | 0.0058 | 0.0084 | 0.0084 | 0.0081 | 0.0106 | 0.0106 | 0.0081 | 0.0106 | 0.0106 | 0.0100 |
| $c_6$ | 0.0007 | 0.0001 | 0.0002 | 0.0001 | 0.0002 | 0.0002 | 0.0000 | 0.0001 | 0.0001 | 0.0000 | 0.0001 | 0.0001 | 0.0004 |
| $c_7$ | 0.0007 | 0.0004 | 0.0005 | 0.0100 | 0.0046 | 0.0149 | 0.0138 | 0.0056 | 0.0187 | 0.0138 | 0.0056 | 0.0187 | 0.0007 |
| $c_8$ | 0.0103 | 0.0046 | 0.0151 | 0.0100 | 0.0150 | 0.0046 | 0.0000 | 0.0004 | 0.0004 | 0.0000 | 0.0004 | 0.0004 | 0.0007 |
| $c_9$ | 0.0103 | 0.0150 | 0.0046 | 0.0001 | 0.0005 | 0.0005 | 0.0138 | 0.0186 | 0.0056 | 0.0138 | 0.0186 | 0.0056 | 0.0181 |
| $c_{10}$ | −0.0022 | −0.0002 | −0.0002 | −0.0001 | −0.0001 | 0.0000 | −0.0003 | −0.0002 | −0.0002 | −0.0003 | −0.0002 | −0.0002 | −0.0002 |
| $c_{11}$ | −0.0008 | −0.0002 | −0.0001 | −0.0001 | 0.0000 | −0.0001 | −0.0002 | −0.0002 | −0.0002 | −0.0002 | −0.0002 | −0.0002 | −0.0002 |
| $c_{12}$ | −0.0008 | −0.0001 | −0.0001 | −0.0002 | −0.0002 | −0.0002 | −0.0003 | −0.0002 | −0.0002 | −0.0003 | −0.0002 | −0.0002 | 0.0004 |
| $c_{13}$ | −0.0008 | 0.0000 | 0.0004 | 0.0168 | 0.0078 | 0.0077 | −0.0003 | −0.0001 | 0.0003 | −0.0003 | −0.0001 | 0.0003 | −0.0002 |
| $c_{14}$ | −0.0008 | 0.0004 | 0.0000 | −0.0001 | 0.0005 | 0.0000 | 0.0230 | 0.0094 | 0.0094 | 0.0230 | 0.0094 | 0.0094 | 0.0009 |
| $c_{15}$ | 0.0166 | 0.0077 | 0.0078 | −0.0001 | 0.0005 | 0.0000 | −0.0003 | 0.0003 | −0.0001 | −0.0003 | 0.0003 | 0.0000 | 0.0009 |
| $c_{16}$ | 0.0021 | −0.0001 | −0.0004 | −0.0005 | −0.0004 | −0.0004 | −0.0003 | −0.0001 | −0.0004 | −0.0003 | −0.0001 | −0.0004 | 0.0000 |
| $c_{17}$ | 0.0003 | −0.0002 | −0.0002 | −0.0005 | −0.0002 | −0.0002 | −0.0009 | −0.0003 | −0.0003 | −0.0009 | −0.0003 | −0.0003 | −0.0004 |
| $c_{18}$ | 0.0003 | −0.0004 | −0.0004 | −0.0003 | −0.0004 | −0.0001 | −0.0003 | −0.0004 | −0.0001 | −0.0003 | −0.0004 | −0.0001 | −0.0004 |
| $c_{19}$ | 0.0021 | −0.0004 | −0.0001 | −0.0003 | −0.0001 | −0.0004 | −0.0009 | −0.0005 | −0.0005 | −0.0009 | −0.0005 | −0.0005 | −0.0004 |
| $c_{20}$ | −0.0032 | −0.0002 | −0.0002 | −0.0004 | −0.0002 | −0.0002 | −−0.0005 | −0.0002 | −0.0002 | −0.0005 | −0.0002 | −0.0002 | 0.0000 |

| | $N_3CPa2$ | $N_3CPa3$ | $N_3CPb1$ | $N_3CPb2$ | $N_3CPb3$ | $N_3CPc1$ | $N_3CPc2$ | $N_3CPc3$ | $N_3CPd1$ | $N_3CPd2$ | $N_3CPd3$ |
|---|---|---|---|---|---|---|---|---|---|---|---|
| $c_1$ | 0.0807 | 0.0853 | 0.0963 | 0.0805 | 0.0858 | 0.0959 | 0.0809 | 0.0843 | 0.0950 | 0.0836 | 0.0835 |
| $c_2$ | 0.0951 | 0.0830 | 0.0839 | 0.0949 | 0.0637 | 0.0844 | 0.0948 | 0.0828 | 0.0798 | 0.0941 | 0.0830 |
| $c_3$ | 0.0796 | 0.0941 | 0.0848 | 0.0807 | 0.1261 | 0.0842 | 0.0790 | 0.0941 | 0.0742 | 0.0729 | 0.0730 |
| $c_4$ | 0.0743 | 0.0730 | 0.0713 | 0.0743 | 0.0712 | 0.0712 | 0.0743 | 0.0731 | 0.0792 | 0.0838 | 0.0941 |
| $c_5$ | 0.0100 | 0.0099 | 0.0101 | 0.0101 | 0.0101 | 0.0100 | 0.0100 | 0.0099 | 0.0099 | 0.0098 | 0.0098 |
| $c_6$ | 0.0008 | 0.0008 | 0.0004 | 0.0008 | 0.0007 | 0.0004 | 0.0008 | 0.0008 | 0.0005 | 0.0008 | 0.0008 |
| $c_7$ | 0.0005 | 0.0008 | 0.0007 | 0.0005 | 0.0005 | 0.0007 | 0.0005 | 0.0008 | 0.0008 | 0.0005 | 0.0008 |
| $c_8$ | 0.0008 | 0.0005 | 0.0007 | 0.0008 | 0.0005 | 0.0007 | 0.0008 | 0.0005 | 0.0179 | 0.0178 | 0.0178 |
| $c_9$ | 0.0181 | 0.0179 | 0.0183 | 0.0182 | 0.0182 | 0.0181 | 0.0180 | 0.0179 | 0.0008 | 0.0008 | 0.0005 |
| $c_{10}$ | −0.0002 | −0.0002 | −0.0002 | −0.0002 | −0.0002 | −0.0002 | −0.0002 | −0.0002 | −0.0002 | −0.0002 | −0.0002 |

(Continued.)

**Table 2.** (*Continued.*)

| | | | | | | | | | | | |
|---|---|---|---|---|---|---|---|---|---|---|---|
| $c_{11}$ | −0.0002 | −0.0002 | −0.0002 | −0.0002 | −0.0003 | −0.0002 | −0.0002 | −0.0002 | 0.0005 | 0.0010 | 0.0010 |
| $c_{12}$ | 0.0010 | 0.0009 | 0.0003 | 0.0009 | 0.0009 | 0.0004 | 0.0010 | 0.0010 | −0.0002 | −0.0002 | −0.0002 |
| $c_{13}$ | −0.0002 | −0.0002 | −0.0002 | −0.0002 | −0.0002 | −0.0002 | −0.0002 | −0.0002 | 0.0010 | 0.0005 | 0.0010 |
| $c_{14}$ | 0.0005 | 0.0009 | 0.0009 | 0.0004 | 0.0006 | 0.0009 | 0.0005 | 0.0010 | −0.0002 | −0.0002 | −0.0002 |
| $c_{15}$ | 0.0010 | 0.0004 | 0.0009 | 0.0009 | 0.0004 | 0.0009 | 0.0010 | 0.0004 | 0.0010 | 0.0010 | 0.0005 |
| $c_{16}$ | 0.0000 | 0.0000 | 0.0000 | 0.0000 | 0.0000 | 0.0000 | 0.0000 | 0.0000 | −0.0004 | −0.0004 | −0.0004 |
| $c_{17}$ | −0.0004 | −0.0004 | −0.0004 | −0.0004 | −0.0004 | −0.0004 | −0.0004 | −0.0004 | −0.0004 | −0.0004 | −0.0004 |
| $c_{18}$ | −0.0004 | −0.0004 | −0.0004 | −0.0004 | −0.0005 | −0.0004 | −0.0004 | −0.0004 | −0.0004 | −0.0004 | −0.0004 |
| $c_{19}$ | −0.0004 | −0.0004 | −0.0004 | −0.0004 | −0.0003 | −0.0004 | −0.0004 | −0.0004 | 0.0000 | 0.0000 | 0.0000 |
| $c_{20}$ | 0.0000 | 0.0000 | 0.0000 | 0.0000 | 0.0000 | 0.0000 | 0.0000 | 0.0000 | 0.0000 | 0.0000 | 0.0000 |

*et al.* [54] are given in parenthesis in figure 1. The four types of rings, denoted A, B, C and D, are shown in figure 2. When compared with the NICS(0) value of $N_0CP1$, ring A shows relatively weak aromaticity, while ring B possesses very significant aromatic character. The negative values correspond to the proposed local aromatic character predicted by our BRE and CRE methods. However, the NICS(0) values for ring A and B vary widely, ranging from −2.8 to −12.5. That is to say, the NICS(0) value ascribed to the B ring is four times larger than that ascribed to the A ring. In the case of $N_1CP2$, the NICS index predicts antiaromatic character for the A ring because the corresponding NICS(0) values are positive, but that contradicts our BRE and CRE predictions. In the case of $N_1CP5$, the four pyrrolic rings exhibit negative NICS(0) values. The signs of the NICS(0) values of the four pyrrolic rings accord with our BRE and CRE values. However, the variations in local aromaticity which were predicted to exist between the A, B and C rings were quite large. As can be seen, in some cases the local aromaticity of $N_0CP1$, $N_1CP2$ and $N_1CP5$ as predicted by the NICS(0) indices does not correspond to the changes in local aromaticity shown by the BRE and CRE indices.

## 3.3. The RC results

The RC effects of the NCP rings have not been reported to date. RC has proven to a very reliable tool for determining the extent of aromaticity. In general, in monocyclic aromatic compounds, magnetically induced diatropic and paratropic RC are associated with their aromaticity and antiaromaticity, respectively [55]. The Hückel–London theory was used here to derive RC patterns of the above compounds [36–41]. In order to better understand the role of the diatropic and paratropic contributions of the different circuits to the RC, we investigated the trend of the magnetic properties of the different circuits. As shown in figure 3, there are a possible 20 circuits in these π-systems. The calculated circuit current (CC) values for these compounds are given in electronic supplementary material, table S2. In general, a positive or negative CC value indicates a diatropic or a paratropic contribution, respectively, to a given circuit. As can be seen in electronic supplementary material, table S3, all circuits in $N_0CP1$ and $N_0CP2$, exhibit positive CC values; thus we can predict that all circuits make a diatropic contribution to the molecule as a whole. Of all the circuits, circuit $c_{14}$ in $N_0CP1$ and circuit $c_{10}$ in $N_0CP2$, which are related to their macrocyclic conjugation pathways, exhibit the largest positive CC values. Thus we can predict that circuit $c_{14}$ in $N_0CP1$ and circuit $c_{10}$ in $N_0CP2$, which are both 18π annulene rings, are respectively, the dominant contributors to the diatropicity of $N_0CP1$ and of $N_0CP2$. In electronic supplementary material, table S3, for each compound, the value of the circuit with the largest predicted diatropicity is shown in bold. As shown there, in the first group and in the second group of isomers, the 18π- and the formally 17π-electron macrocyclic conjugation pathways, respectively, exhibited the largest CC values. Thus, we can conclude that the 18π macrocyclic conjugation pathway is the main source of the diatropicity of the first group of isomers, and the formally 17π-electron macrocyclic conjugation pathway is the main source of diatropicity in the second group of isomers. The RC flow through a given ring can be calculated as the sum of the contributions of a given circuit's CC values. The direction and intensity of the RCs of $N_0CP1$, $N_0CP2$ and $N_1CP2$ are given in figure 4, which also shows that diatropic currents flow counterclockwise and paratropic currents flow clockwise. The RCs of the remaining compounds are shown in electronic supplementary material, figure S2. As can be seen from figure 4 and electronic supplementary material, figure S2, all compounds were predicted to sustain a diatropic RC along the molecular perimeter. $N_0CP1$ and $N_0CP2$ possess the strongest diatropic RCs among all the molecules we investigated here. With the increase in the number of the *confused* pyrrole rings in the molecule, the

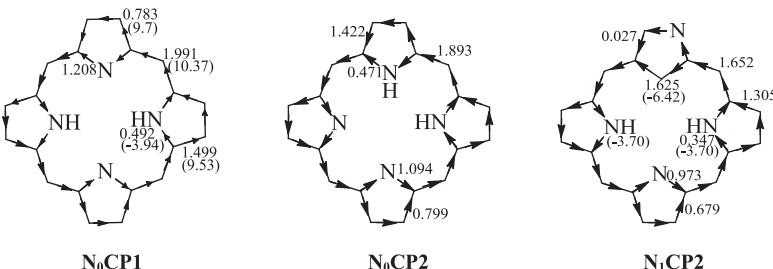

**Figure 4.** The RC values of compounds **N₀CP1**, **N₀CP2** and **N₁CP2**. All in units of that for benzene ($I_0$). The chemical shift values of **N₀CP1** and **N₁CP2** are given in parentheses.

strength of the RCs passing through the $C_\alpha$–$C_{meso}$ bonds gradually decreases. The diatropic RC induced along the macrocycle pathway is divided when it passes through each pyrrole ring. The four pyrrole rings sustain diatropic RCs of their own. The RCs partially cancel each other out on the inner section of the pyrrole rings due to both currents flowing in opposite directions. For **N₀CP1**, regarding its non-protonated pyrrole ring, the RC strength through the $\beta$-position is 0.783 $I_0$, whereas 1.208 $I_0$ flows via the inner section. Regarding its protonated pyrrole ring, the corresponding RC strengths are 1.499 $I_0$ and 0.492 $I_0$, respectively. In **N₀CP1** and **N₀CP2**, the strength of the RC passing through the protonated nitrogen atom is weaker than that passing through the $\beta$-position of the same pyrrolic ring, whereas in the non-protonated pyrrole rings, the RC passing through the non-protonated nitrogen atom is stronger than that passing through the $\beta$-position of that same pyrrolic ring. This is the commonly accepted main current pathway of porphyrins [4].

It is worth noting that in all NCP isomers, the strength of the RC passing through the normal pyrrolic ring section is similar to that of **N₀CP1** and **N₀CP2**. However, the confused pyrrolic ring section is different from that of the non-confused pyrrolic ring. In the confused pyrrolic rings, the RC passing through the inner carbon atom is always stronger than that passing through the $\beta$-position of the same pyrrole ring, whether or not the $\beta$-position is located at the protonated nitrogen or non-protonated nitrogen atom. This is the main difference between **N₀CP1** and **N₀CP2** on the one hand, and the NCP isomers on the other. The experimental value of $^1$H NMR spectra for **N₀CP1** is given in figure 4 in parenthesis [56]. As shown in that figure, the greatest downfield chemical shift of *meso*-protons (10.37 ppm) is in good agreement with their strong diamagnetic RC value (1.991 $I_0$), and the $\beta$-position's chemical shift (9.53 ppm) is also in good agreement with the weakened RC value (0.783 $I_0$). The high upfield chemical shift value (−3.94 ppm) of protonated nitrogen is also in good agreement with their weak diamagnetic RC value (0.492 $I_0$). In the NCP isomers, due to the presence of one or more confused pyrrole rings, there is weaker diatropic RC at the $\beta$-positions than the RC in the normal pyrrole ring(s) within the same molecule. In the case of **N₁CP2**, the 11 peripheral protons resonated between $\delta = 10.3$ and 9.2 ppm in $CDCl_3$ [57]. The downfield chemical shift is in good agreement with their diatropic RCs. The upfield shift of the inner NH signal could be explained by the weak diamagnetic RC, similar to those of **N₀CP1**. The large RC of the inner CH section of the confused pyrrole ring is not in agreement with the observed chemical shifts.

## 4. Conclusion

The positioning of the nitrogen atoms in NCP isomers has a significant effect on the global aromaticity and diatropicity. The presence of the nitrogen atoms on the periphery of the ring results in increased distance between the nitrogen atoms and this serves to disrupt the full conjugation around the macrocycle, which in turn causes these systems to slowly become increasingly less stable. We have seen that aromaticity and magnetropicity originate from two different sources. Based on the BRE and CRE results, we would predict that the pyrrolic rings would continue to exhibit the largest local aromaticity and would continue to be the main source of global aromaticity; however, based on the RC results, we would predict the macrocyclic conjugation pathway to be the main source of diatropicity. This is so because the origin of the aromaticity arising in polycyclic compounds is not always the same as the origin of the magnetropicity. Therefore, if our results are correct, the strength of the RC may not always reflect correctly the degree of aromaticity. The diatropic RC induced along the macrocyclic pathway divides at the pyrroles where it then follows both an outer and an inner

route. In the confused pyrrole rings of the NCP isomers, the strengths of the diatropic RC passing through the $\beta$-positions is always weaker than that passing through the inner routes. Knowing that this is the structure of these RC intensities could be useful in designing or predicting experimental work with the new NCP systems.

Data accessibility. Additional data are available in the electronic supplementary material.

Authors' contributions. A.K. carried out the design of the study, participated in data analysis and wrote the manuscript. M.T. performed the computations and gave valuable suggestions on the writing of the manuscript. All authors gave final approval for publication.

Competing interests. We declare we have no competing interests.

Funding. This work was supported by the National Natural Science Foundation of China (grant no. 21662033).

Acknowledgements. We thank Dr Liu Chen-Jiang (The College of Chemistry and Chemical Engineering, Xinjiang University, Urumqi, China) for checking the language of this paper.

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
