## [Reviewer comments · Royal Society Open Science]

Review History

RSOS-200069.R0 (Original submission)

Review form: Reviewer 1

Is the manuscript scientifically sound in its present form?

Yes

Are the interpretations and conclusions justified by the results?

No

Is the language acceptable?

No

Do you have any ethical concerns with this paper?

No

Have you any concerns about statistical analyses in this paper?

No

Recommendation?

Major revision is needed (please make suggestions in comments)

Comments to the Author(s)

N-confused porphyrins (NCPs) are indeed very interesting molecules particularly in the orientation of the N-lone pairs. The authors need to seriously work on the presentation of the manuscript. Apart from the academic comments mentioned below, they indeed require additional efforts to improve the manuscript.

Scientific comments:

(a) The conformations and the molecular symmetry of the NCPs need to be clearly mentioned as Fig. 1 provides little idea of the 3D-structure.

(b) The authors don't mention the level of calculations like the functionals or the basis-sets used. Please clearly mention and also mention if frequency calculations are performed to determine that these structures are not Saddle-points.

(c) The conclusions should not be written as bullets and they should write their conclusions more carefully. Also, the first conclusion is trivial - NCPs are known to be less stable than porphyrins. Therefore, please remove this.

(D) The authors should also report the NICS_{zzz}(1) as the ring currents are mostly above the planes. NICS(0) are more indications of the sigma-delocalizations.

(E) The following articles should be discussed in the context of aromaticity of molecules and clusters: (1) "Quantifying Aromaticity at the Molecular and Supramolecular Limits: Comparing Homonuclear, Heteronuclear and H-Bonded Systems", *J. Chem. Theory Comput.*, 2, 30. (2006). (2) "Stable transition metal complexes of an all-metal antiaromatic molecule (Al₄Li₄): Role of Complexations", *J. Am. Chem. Soc.* 127, 3496 (2005). (3) "Rationalization of pi-sigma (anti)aromaticity in all-metal molecular cluster", *J. Chem. Theory & Comput.*, 1, 824. (2005).

Review form: Reviewer 2

Is the manuscript scientifically sound in its present form?

Yes

Are the interpretations and conclusions justified by the results?

Yes

Is the language acceptable?

Yes

Do you have any ethical concerns with this paper?

No

Have you any concerns about statistical analyses in this paper?

No

Recommendation?

Accept with minor revision (please list in comments)

Comments to the Author(s)

Since the ring current strengths have been obtained using the Huckel-London model, some care should be used to attribute a general validity to the obtained results. When ab-initio methods are

employed to calculate the RC strengths, very accurate results are usually obtained which can be directly related to the degree of aromaticity. Therefore, a bit of prudence should be used when describing the RC result, as for example in sentence like: "the strength of the RC does not always reflect correctly the degree of aromaticity", which does not seem always valid, i.e., in absolute. Otherwise the paper is very well written and recommended for publication after a very, very minor revision as described above.

Decision letter (RSOS-200069.R0)

02-Apr-2020

Dear Professor Kerim:

Title: A Study on the Aromaticity and Magnetic Properties of N-Confused Porphyrins
Manuscript ID: RSOS-200069

The editor assigned to your manuscript has now received comments from reviewers. We would like you to revise your paper in accordance with the referee and Subject Editor suggestions which can be found below (not including confidential reports to the Editor). Please note this decision does not guarantee eventual acceptance.

Please submit your revised paper before 25-Apr-2020. Please note that the revision deadline will expire at 00.00am on this date. If we do not hear from you within this time then it will be assumed that the paper has been withdrawn. In exceptional circumstances, extensions may be possible if agreed with the Editorial Office in advance. We do not allow multiple rounds of revision so we urge you to make every effort to fully address all of the comments at this stage. If deemed necessary by the Editors, your manuscript will be sent back to one or more of the original reviewers for assessment. If the original reviewers are not available we may invite new reviewers.

On behalf of the Subject Editor Professor Anthony Stace and the Associate Editor Dr Debashree Ghosh.

RSC Associate Editor:
Comments to the Author:
(There are no comments.)

RSC Subject Editor:
Comments to the Author:
(There are no comments.)

Reviewers' Comments to Author:
Reviewer: 1

Comments to the Author(s)
N-confused porphyrins (NCPs) are indeed very interesting molecules particularly in the orientation of the N-lone pairs. The authors need to seriously work on the presentation of the manuscript. Apart from the academic comments mentioned below, they indeed require additional efforts to improve the manuscript.

Scientific comments:

(a) The conformations and the molecular symmetry of the NCPs need to be clearly mentioned as Fig. 1 provides little idea of the 3D-structure.

(b) The authors don't mention the level of calculations like the functionals or the basis-sets used. Please clearly mention and also mention if frequency calculations are performed to determine that these structures are not Saddle-points.

(c) The conclusions should not be written as bullets and they should write their conclusions more carefully. Also, the first conclusion is trivial - NCps are known to be less stable than porphyrins. Therefore, please remove this.

(D) The authors should also report the NICS_{zz}(1) as the ring currents are mostly above the planes. NICS(0) are more indications of the sigma-delocalizations.

(E) The following articles should be discussed in the context of aromaticity of molecules and clusters: (1) "Quantifying Aromaticity at the Molecular and Supramolecular Limits: Comparing Homonuclear, Heteronuclear and H-Bonded Systems", J. Chem. Theory Comput. , 2, 30. (2006). (2) "Stable transition metal complexes of an all-metal antiaromatic molecule (Al₄Li₄):Role of Complexations", J. Am. Chem.Soc. 127, 3496 (2005). (3) "Rationalization of pi-sigma (anti)aromaticity in all-metal molecular cluster", J. Chem. Theory & Compt., 1, 824. (2005).

Reviewer: 2

Comments to the Author(s)

Since the ring current strengths have been obtained using the Huckel-London model, some care should be used to attribute a general validity to the obtained results. When ab-initio methods are employed to calculate the RC strengths, very accurate results are usually obtained which can be directly related to the degree of aromaticity. Therefore, a bit of prudence should be used when describing the RC result, as for example in sentence like: "the strength of the RC does not always reflect correctly the degree of aromaticity", which does not seem always valid, i.e., in absolute. Otherwise the paper is very well written and recommended for publication after a very, very minor revision as described above.

Author's Response to Decision Letter for (RSOS-200069.R0)

See Appendix A.

Decision letter (RSOS-200069.R1)

Dear Professor Kerim:

Title: A Study on the Aromaticity and Magnetic Properties of N-Confused Porphyrins
Manuscript ID: RSOS-200069.R1

Thank you for submitting the above manuscript to Royal Society Open Science. On behalf of the Editors and the Royal Society of Chemistry, I am pleased to inform you that your manuscript will be accepted for publication in Royal Society Open Science subject to minor revision in accordance with the referee suggestions. Please find the reviewers' comments at the end of this email.

The reviewers and handling editors have recommended publication, but also suggest some minor revisions to your manuscript. Therefore, I invite you to respond to the comments and revise your manuscript.

Because the schedule for publication is very tight, it is a condition of publication that you submit the revised version of your manuscript before 15-May-2020. Please note that the revision deadline will expire at 00.00am on this date. If you do not think you will be able to meet this date please let me know immediately.

When submitting your revised manuscript, you will be able to respond to the comments made by the referees and upload a file "Response to Referees" in "Section 6 - File Upload". You can use this to document any changes you make to the original manuscript. In order to expedite the

processing of the revised manuscript, please be as specific as possible in your response to the referees.

Kind regards,
Dr Laura Smith
Publishing Editor, Journals

On behalf of the Subject Editor Professor Anthony Stace and the Associate Editor Dr Debashree Ghosh.

RSC Associate Editor

Comments to the Author:

The authors seem to have answered all the referee queries and modifies their manuscript accordingly. However, referee 1 asked for the conclusion to not be written as a bullet point which is also something I think is appropriate. This has not been addressed by the author. Apart from this minor change I believe the manuscript can be accepted.

Reviewer comments to Author:

Author's Response to Decision Letter for (RSOS-200069.R1)

See Appendix B.

Decision letter (RSOS-200069.R2)

Dear Professor Kerim:

Title: A Study on the Aromaticity and Magnetic Properties of N-Confused Porphyrins
Manuscript ID: RSOS-200069.R2

It is a pleasure to accept your manuscript in its current form for publication in Royal Society Open Science. The chemistry content of Royal Society Open Science is published in collaboration with the Royal Society of Chemistry.

On behalf of the Subject Editor Professor Anthony Stace and the Associate Editor Dr Debashree Ghosh.

RSC Associate Editor
Comments to the Author:
(There are no comments.)

Reviewer(s)' Comments to Author:

Appendix A

Dear Editor:

Thank you for your useful comments and suggestions on the structure of our manuscript. We have modified the manuscript accordingly, and detailed corrections are listed below point by point.

Title: A Study on the Aromaticity and Magnetic Properties of N-Confused Porphyrins

Manuscript ID: RSOS-200069

Correspondence Author: Ablikim Kerim

Authors: Maimaitijiang Tuersun and Ablikim Kerim*

Reviewers' Comments to Author:

Reviewer: 1

Comments to the Author(s)

N-confused porphyrins (NCPs) are indeed very interesting molecules particularly in the orientation of the N-lone pairs. The authors need to seriously work on the presentation of the manuscript. Apart from the academic comments mentioned below, they indeed require additional efforts to improve the manuscript.

Scientific comments:

(a) The conformations and the molecular symmetry of the NCPs need to be clearly mentioned as Fig. 1 provides little idea of the 3D-structure.

a) Response:

In Fig. 1, we showed the 37 kinds of NCP isomers. In the structure of these compounds, we have provided both of the BRE values of the different π bonds and the main conjugation pathways are shown in bold. In addition, the NICS(0) values for the individual rings of N₀CP1, N₁CP2, and N₁CP5 are given in parentheses. When these isomers are displayed in a 3D format instead of 2D, the hydrogen atoms take up a great deal of space. If we provide the BRE values inside a 3D-structure, the BRE values of the different bonds and of the main conjugation pathways become difficult to read, and the entire figure might take several pages to depict. We fear that our article would become overly long.

For this reason Furuta et al., (perhaps even in their original paper), provided the NCP isomers in 2D (plane) format. Such a format seems to be fairly common. Thus, we also chose this method.

(b) The authors don't mention the level of calculations like the functionals or the basis-sets used. Please clearly mention and also mention if frequency calculations are performed to determine that these structures are not Saddle-points.

b) Response:

We believe that Referee 1 has misunderstood some aspects of our paper. All our calculation methods are based on simple Hückel molecular orbital theory. This method does not provide

any information about the frequency calculations (Saddle-points) of the systems.

TRE is defined as the difference between the π -electron energy of a conjugated molecule and its hypothetical acyclic reference structure (matching polynomials). The hypothetical acyclic reference structure can be obtained using Sachs theorem. Regarding the matching polynomials, please see the following reference.

K.Balasubramanian, Exhaustive generation and analytical expressions of matching polynomials of fullerenes C₂₀-C₅₀. J. Chem. Inf. Comput. Sci.,1994, 34, 421-427.

In the “Methods of Calculation” section, We have included the method of our calculations:

“The TRE, BRE, CRE, and RC indices can be defined graph-theoretically within the framework of simple Hückel molecular orbital theory.”

Do you think this is not enough detail?

(c) The conclusions should not be written as bullets and they should write their conclusions more carefully. Also, the first conclusion is trivial - NCPs are known to be less stable than porphyrins. Therefore, please remove this.

c) Response:

We agree with this suggestion and according to the comments of Reviewer 1, we have revised the concluding section of our paper.

Previous version:

On the basis of the data presented, the following conclusions can be drawn:

- 1) According to the TRE results, we can predict that all the NCPs are less stable than normal porphyrins due to the non-conformity of the NCP isomers to the TCS rule.
- 2) The aromaticity and magnetropicity originate from two different sources. Based on the BRE and CRE results, we predict that the pyrrolic rings will continue to exhibit the largest local aromaticity and will continue to be the main source of global aromaticity. However, based on the RC results, we predict the macrocyclic conjugation pathway to be the main source of diatropicity. This is so because the origin of the aromaticity arising in polycyclic compounds is not always same as the origin of the magnetropicity. The strength of the RC does not always reflect correctly the degree of aromaticity.
- 3) In the confused pyrrole rings of the NCP isomers, the diatropic RC passing through the β -positions is always weaker than that passing through the inner sections. This is the key difference between the confused pyrrolic rings and the normal pyrrolic rings in the NCP molecules.

New version:

Upon analysis, the data from this study allows several conclusions to be drawn.

1. The positioning of the nitrogen atoms in NCP isomers has a significant effect on the global aromaticity and diatropicity. The presence of the nitrogen atoms on the periphery of the ring results in increased distance between the nitrogen atoms and this serves to disrupt the full conjugation around the macrocycle, which in turn causes these systems to slowly become increasingly less stable.
2. We have seen that aromaticity and magnetropicity originate from two different sources. Based on the BRE and CRE results, we would predict that the pyrrolic rings would continue

to exhibit the largest local aromaticity and would continue to be the main source of global aromaticity; however, based on the RC results, we would predict the macrocyclic conjugation pathway to be the main source of diatropicity. This is so because the origin of the aromaticity arising in polycyclic compounds is not always the same as the origin of the magnetropicity. Therefore, if our results are correct, the strength of the RC may not always reflect correctly the degree of aromaticity.

3. In the confused pyrrole rings of the NCP isomers, the diatropic RC passing through the β -positions is always weaker than that passing through the inner sections. Knowing that this is the structure of these RC intensities could be useful in designing or predicting experimental work with the new NCP systems.

(D) The authors should also report the NICS_{zz}(1) as the ring currents are mostly above the planes. NICS(0) are more indications of the sigma-delocalizations.

d) Response:

We agree, and in accordance with this suggestion of Reviewer 1, we have revised of our paper and added related information:

Previous expression:

The nucleus-independent chemical shift (NICS) index is one of the most widely used magnetic criteria of aromaticity, and several variations of the NICS index have been proposed.

New expression:

The nucleus-independent chemical shift (NICS) index is one of the most widely used magnetic criteria of aromaticity. The index is defined as the negative value of the absolute magnetic shielding computed at ring center NICS(0), and 1 Å above the center of the ring NICS(1) and its zz-tensor component NICS(1)_{zz}, where the z-axis is a normal to the ring plane.

(E) The following articles should be discussed in the context of aromaticity of molecules and clusters: (1) “Quantifying Aromaticity at the Molecular and Supramolecular Limits: Comparing Homonuclear, Heteronuclear and H-Bonded Systems”, J. Chem. Theory Comput. , 2, 30. (2006). (2) “Stable transition metal complexes of an all-metal antiaromatic molecule (Al₄Li₄):Role of Complexations”, J. Am. Chem.Soc. 127, 3496 (2005). (3) “Rationalization of pi-sigma (anti)aromaticity in all-metal molecular cluster”, J. Chem. Theory & Compt., 1, 824. (2005).

e) Response:

We have now added these three references at the relevant position in our paper:

Reviewer: 2

Comments to the Author(s)

Since the ring current strengths have been obtained using the Huckel-London model, some care should be used to attribute a general validity to the obtained results. When ab-initio methods are employed to calculate the RC strengths, very accurate results are usually obtained which can be directly related to the degree of aromaticity. Therefore, a bit of prudence should be used when describing the RC result, as for example in sentence like: “the strength of the RC does not always

reflect correctly the degree of aromaticity”, which does not seem always valid, i.e., in absolute. Otherwise the paper is very well written and recommended for publication after a very, very minor revision as described above.

Response:

We agree with this suggestion and after careful consideration of the comments of Reviewer 2, We have softened the tone of that sentence in our conclusion.

Previous expression:

This is so because the origin of the aromaticity arising in polycyclic compounds is not always same as the origin of the magnetropicity. The strength of the RC does not always reflect correctly the degree of aromaticity.

New expression:

This is so because the origin of the aromaticity arising in polycyclic compounds is not always the same as the origin of the magnetropicity. Therefore, if our results are correct, the strength of the RC may not always reflect correctly the degree of aromaticity.

If you find any further problems with this paper or ways in which it can be improved, please do inform us and we will gladly make those revisions.

Sincerely,

The Correspondence author: Ablikim Kerim

All authors: Maimaitijiang Tuersun and Ablikim Kerim

College of Chemistry and Chemical Engineering, Xinjiang University, Urumqi 830046, China

E-mail address: ablikim.kerim@163.com

Tel: 00 86-991-8582809

Fax: 0086-991-8582809

2020-04-16

Appendix B

Dear Editor:

Thank you for your useful comments and suggestions on the structure of our manuscript. We have modified the manuscript accordingly, and detailed corrections are listed below point by point.

Title: A Study on the Aromaticity and Magnetic Properties of N-Confused Porphyrins

Manuscript ID: RSOS-200069.R1

Correspondence Author: Ablikim Kerim

Authors: Maimaitijiang Tuersun and Ablikim Kerim*

Comments to the Author:

The authors seem to have answered all the referee queries and modifies their manuscript accordingly. However, referee 1 asked for the conclusion to not be written as a bullet point which is also something I think is appropriate. This has not been addressed by the author. Apart from this minor change I believe the manuscript can be accepted.

Response:

We agree with this suggestion and in accordance with the comments of this Reviewer, we have revised the concluding section of our paper.

Our original version:

On the basis of the data presented, the following conclusions can be drawn:

- 1) According to the TRE results, we can predict that all the NCPs are less stable than normal porphyrins due to the non-conformity of the NCP isomers to the TCS rule.
- 2) The aromaticity and magnetropicity originate from two different sources. Based on the BRE and CRE results, we predict that the pyrrolic rings will continue to exhibit the largest local aromaticity and will continue to be the main source of global aromaticity. However, based on the RC results, we predict the macrocyclic conjugation pathway to be the main source of diatropicity. This is so because the origin of the aromaticity arising in polycyclic compounds is not always same as the origin of the magnetropicity. The strength of the RC does not always reflect correctly the degree of aromaticity.
- 3) In the confused pyrrole rings of the NCP isomers, the diatropic RC passing through the β -positions is always weaker than that passing through the inner sections. This is the key difference between the confused pyrrolic rings and the normal pyrrolic rings in the NCP molecules.

Our second version:

Upon analysis, the data from this study allows several conclusions to be drawn.

1. The positioning of the nitrogen atoms in NCP isomers has a significant effect on the global aromaticity and diatropicity. The presence of the nitrogen atoms on the periphery of the ring results in increased distance between the nitrogen atoms and this serves to disrupt the full conjugation around the macrocycle, which in turn causes these systems to slowly become increasingly less stable.
2. We have seen that aromaticity and magnetropicity originate from two different sources. Based on the BRE and CRE results, we would predict that the pyrrolic rings would continue

to exhibit the largest local aromaticity and would continue to be the main source of global aromaticity; however, based on the RC results, we would predict the macrocyclic conjugation pathway to be the main source of diatropicity. This is so because the origin of the aromaticity arising in polycyclic compounds is not always the same as the origin of the magnetropicity. Therefore, if our results are correct, the strength of the RC may not always reflect correctly the degree of aromaticity.

3. In the confused pyrrole rings of the NCP isomers, the diatropic RC passing through the β -positions is always weaker than that passing through the inner sections. Knowing that this is the structure of these RC intensities could be useful in designing or predicting experimental work with the new NCP systems.

Our present version:

4. Conclusions

The positioning of the nitrogen atoms in NCP isomers has a significant effect on the global aromaticity and diatropicity. The presence of the nitrogen atoms on the periphery of the ring results in increased distance between the nitrogen atoms and this serves to disrupt the full conjugation around the macrocycle, which in turn causes these systems to slowly become increasingly less stable. We have seen that aromaticity and magnetropicity originate from two different sources. Based on the BRE and CRE results, we would predict that the pyrrolic rings would continue to exhibit the largest local aromaticity and would continue to be the main source of global aromaticity; however, based on the RC results, we would predict the macrocyclic conjugation pathway to be the main source of diatropicity. This is so because the origin of the aromaticity arising in polycyclic compounds is not always the same as the origin of the magnetropicity. Therefore, if our results are correct, the strength of the RC may not always reflect correctly the degree of aromaticity. The diatropic RC induced along the macrocyclic pathway divides at the pyrroles where it then follows both an outer and an inner route. In the confused pyrrole rings of the NCP isomers, the strengths of the diatropic RC passing through the β -positions is always weaker than that passing through the inner routes. Knowing that this is the structure of these RC intensities could be useful in designing or predicting experimental work with the new NCP systems.

If you find any further problems with this paper or ways in which it can be improved, please do inform us and we will gladly make those revisions.

Sincerely,

The Correspondence author: Ablikim Kerim

All authors: Maimaitijiang Tuersun and Ablikim Kerim

College of Chemistry and Chemical Engineering, Xinjiang University, Urumqi 830046, China

E-mail address: ablikim.kerim@163.com

Tel: 00 86-991-8582809

Fax: 0086-991-8582809

2020-05-18